# The Metabolic Heterogeneity and Flexibility of Cancer Stem Cells

**DOI:** 10.3390/cancers12102780

**Published:** 2020-09-28

**Authors:** Atsushi Tanabe, Hiroeki Sahara

**Affiliations:** Laboratory of Biology Azabu University School of Veterinary Medicine 1-17-71 Fuchinobe, Chuo-ku, Sagamihara, Kanagawa 252-5201, Japan; at-tanabe@azabu-u.ac.jp

**Keywords:** cancer stem cells, glucose metabolism, mitochondrial metabolism, redox homeostasis, ROS, metastasis

## Abstract

**Simple Summary:**

Cancer stem cells (CSCs) have been shown to be the main cause of therapy resistance and cancer recurrence. An analysis of their biological properties has revealed that CSCs have a particular metabolism that differs from non-CSCs to maintain their stemness properties. In this review, we analyze the flexible metabolic mechanisms of CSCs and highlight the new therapeutics that target CSC metabolism.

**Abstract:**

Numerous findings have indicated that CSCs, which are present at a low frequency inside primary tumors, are the main cause of therapy resistance and cancer recurrence. Although various therapeutic methods targeting CSCs have been attempted for eliminating cancer cells completely, the complicated characteristics of CSCs have hampered such attempts. In analyzing the biological properties of CSCs, it was revealed that CSCs have a peculiar metabolism that is distinct from non-CSCs to maintain their stemness properties. The CSC metabolism involves not only the catabolic and anabolic pathways, but also intracellular signaling, gene expression, and redox balance. In addition, CSCs can reprogram their metabolism to flexibly respond to environmental changes. In this review, we focus on the flexible metabolic mechanisms of CSCs, and highlight the new therapeutics that target CSC metabolism.

## 1. Introduction

Over the years, as the biological properties of cancers become clearer, various therapies have been developed to target them. For example, therapeutic agents that target the vigorous cell growth of cancer cells, DNA replication inhibitors, and cell division inhibitors have shown dramatic effects for tumor regression [1,2,3,4,5]. However, while many therapeutic agents have transient effects, tumors often become refractory and repopulate by acquiring treatment resistance [1,3,5]. Cancer stem cells (CSCs) or tumor-initiating cells, which represent a small population of cells existing inside cancer tissues, are responsible for treatment resistance and the recurrence of cancers [6,7,8,9,10,11,12]. Since the discovery of CSCs in leukemia cells about 30 years ago [7], CSC research has been conducted in various hematological and solid tumors [12]. CSCs can generate cancer cells with different characteristics by dividing unevenly while maintaining a poorly differentiated state [9,11]. It is considered that such pluripotency and self-renewal ability are the reasons for the heterogeneity in cancer tissues, and the cause of treatment resistance and recurrence [8].

A number of molecular markers have been identified to isolate CSCs from primary tumors and experimental tumor models. Although there are molecular markers that are common between tumors of various tissues, such as CD44 and aldehyde dehydrogenase (ALDH), most CSCs express tissue-specific molecular markers [12]. However, emerging evidence suggests that the phenotype of CSCs originating from a tissue is not always constant, but changes in a context-dependent manner. For example, it has been shown that multiple CSC populations with different growth and metastatic traits are present simultaneously within the same tumor [13,14]. At present, the metabolic mechanism of cancer cells is attracting attention as a predisposing factor underlying the diversity of CSCs [15,16,17,18,19,20]. One hundred years ago, when it was discovered that the glucose metabolism of cancer cells was different from that of normal cells, the quantities and categories of metabolites that could be analyzed were limited [21]. However, with the progress of metabolomics and isotope tracing technology in the recent years, it is now possible to investigate not only changes in quantity, but also the details of the flux of metabolites and metabolic heterogeneities within tumors [22,23]. In this review, we describe the details of tumor metabolism revealed in recent studies and discuss the therapeutic potential of the CSC-specific metabolic machinery.

## 2. Tumor Metabolism

The specialized metabolism of tumors was first revealed by Warburg et al. in 1927 [21]. They discovered that cancer cells have a high dependence on glucose, and this property of cancer cells is still being applied in tumor imaging techniques, such as positron emission tomography [24,25]. Over the years, cancer metabolism and the Warburg effect have been treated synonymously, but recent studies have revealed that the Warburg effect is only one aspect of the metabolic mechanisms of tumors [26,27,28,29]. Each tumor activates different metabolic pathways in response to gene mutations and changes in the microenvironment [26,27,28,29].

### 2.1. Glucose Metabolism

In typical glycolysis, 1 mole of glucose is converted to 2 moles of pyruvate by 10 enzymes. Under normal oxygen conditions, the 2 moles of pyruvate are completely oxidized through the mitochondrial tricarboxylic acid (TCA) cycle, and 30 or 32 moles of ATP are produced by the electron transport system (ETS). However, even when there is sufficient oxygen and the mitochondria are functioning normally, cancer cells excrete most of the pyruvate as lactate [21,24,25]. Although this is energetically inefficient, it can be advantageous for cancer cells, because glycolytic intermediates, which are generated in the pre-stage of lactate secretion, are linked to various other metabolic pathways that produce biogenic substances required for cell growth (Figure 1). The pentose phosphate pathway (PPP) is an important pathway for supplying ribose, which is a material for nucleic acids, and NADPH, which plays an important role in maintaining intracellular redox balance [30]. Dihydroxyacetone phosphate, which is produced by the degradation of fructose bisphosphate, is reduced to glycerol-3 phosphate, and becomes a material for cell membranes [22]. 3-Phosphoglycerate is metabolized to serine, then undergoes one-carbon metabolism to become a material for nucleic acids and glutathione (GSH), or is metabolized to S-adenosylmethionine, which is a methyl group donor for proteins and DNA [31,32]. To accelerate glycolysis, cancer cells convert NADH generated in the metabolic process into NAD+ by using it for the reduction of pyruvate without transporting it to the ETS [24]. Lactate produced by the reduction of pyruvate is excreted extracellularly, and it becomes an energy source for other cancer cells and stromal cells [33,34]. Lactate also promotes the metastasis of cancer cells and suppresses the function of immune cells by acidifying the tumor microenvironment [33,34,35].

Many transporters and enzymes involve in glucose uptake and metabolism, e.g., glucose transporters (GLUTs), monocarboxylate transporters, hexokinase 2, lactate dehydrogenase A, and pyruvate dehydrogenase kinase 1, and transcription factors that regulate their expression, e.g., hypoxia inducible factor-1 alpha (HIF-1α) and c-myc, are overexpressed in various cancers [36,37,38,39,40,41,42]. Similarly, it is well-known that in various cancers, gene mutations and dysregulation occur in Kras/mitogen-activated protein kinase (MAPK) and phosphatidylinositol-3 kinase (PI3K)/Akt/mTOR, which are signal transduction pathways related to glycolysis control [26,43]. It is also known that the tumor suppressor gene p53 has a role in suppressing glycolysis by inhibiting GLUTs expression and phosphofructokinase activity [44,45,46,47,48,49], whereas mutant p53 conversely promotes glucose uptake and glycolysis [48,49].

### 2.2. Mitochondrial Metabolism and Oxidative Phosphorylation

Similar to glycolysis, various metabolites produced by mitochondrial metabolism, including the TCA cycle, are required for building cellular components and energy production [50,51,52]. Acetyl-CoA produced by the degradation of citrate is a material for fatty acid synthesis and an acetyl group donor for the histone acetylation reaction [50,51,52]. Oxaloacetate is converted to aspartate by glutamate-oxaloacetate transaminase, and the aspartate also serves as a material for nucleic acids [53]. The main fuel of the TCA cycle is acetyl-CoA that is derived from glucose, but when there are insufficient amounts of available glucose or under conditions of hypoxia, glutamine, other branched-chain amino acids, and acetyl-CoA produced by the β-oxidation of fatty acids are used as fuel in the TCA cycle [28,37,54].

Compared to normal tissues, glycolysis is activated in most cancer cells, but the cancer cells still produce the ATP required for growth and survival via the mitochondrial oxidative phosphorylation (OXPHOS) system [55]. The ETS reaction is essential for generating the proton-driving force necessary for synthesizing large amounts of ATP, and at the same time, the reaction causes the production of reactive oxygen species (ROS), which are harmful to cells. Therefore, mitochondria are equipped with numerous ROS defense mechanisms, and many of them are upregulated in cancer cells [50,51,52]. While excessive amounts of ROS are harmful to cancer cells, it is known that an appropriate amount of ROS plays an important role in signal transduction and the differentiation of cancer cells [52,56]. The various effects of ROS on the phenotype of CSCs will be described later.

Peroxisome proliferator-activated receptor gamma coactivator 1-alpha (PGC-1α), which plays a central role in mitochondrial biosynthesis, is overexpressed in various cancers [57,58]. PGC-1α has been reported to induce cancer growth and distant metastasis in a mitochondrial metabolism-dependent manner [59,60]. c-Myc is known to be involved in mitochondrial biosynthesis and the regulation of the expression of glycolysis-related genes [61]. It has been shown that cancer cells overexpressing c-myc have increased mitochondrial masses and activated OXPHOS [61,62].

It has been reported that mutations of genes encoding enzymes involved in mitochondrial metabolism affect the phenotype of cancer cells. A mutation of isocitrate dehydrogenase 1 and 2 found in glioma produces D-2-hydroxy-glutarate from α-ketoglutarate, which results in the inhibition of DNA and histone methyltransferases, and causes epigenetic dysregulation [55,63,64]. Mutations in succinate dehydrogenase and fumarate hydratase inhibit prolyl hydroxylase activity by accumulating succinate and fumarate in cells, and suppress HIF-1α degradation [55,65].

### 2.3. Amino Acids, Lipids and Other Fuels

Glucose is a raw material that makes up various biological substances, but it contains only carbon, hydrogen, and oxygen. Since glutamine plays an important role as a source of nitrogen in amino acids and nucleic acids, it is known that many cancer cells have a high dependence on glutamine [66,67,68,69,70]. Glutamine uptake and metabolism in cancer cells is regulated by PI3K, Kras, and c-myc [26,43,66,67,71,72,73,74,75]. Tryptophan is an important amino acid for the activation of immune cells, but cancer cells overexpress the tryptophan-metabolizing enzyme IDO, which suppresses immune cell function by depleting tryptophan in the tumor microenvironment [66]. In addition, it has been reported that kynurenine produced by the catabolism of tryptophan enhances the resistance of CSCs to radiation [76]. Acute lymphocytic leukemia is known to have a high dependence on asparagine [66,67,77]. The treatment method of using L-asparaginase to degrade and deplete asparagine in the tumor microenvironment has shown a high therapeutic effect on acute lymphocytic leukemia in children [66,67,77,78].

A large amount of lipids, a component of biological membranes, is required for the rapid cell division of cancer cells. Therefore, the overexpression of fatty acid synthetase and sterol regulatory element-binding protein, which are involved in lipid and cholesterol synthesis, is observed in cancer cells [79]. In addition, it has been reported that the CD36 receptor, which regulates the uptake of lipids, promotes the growth and metastasis of cancer cells [80,81,82]. Synthesized or incorporated lipids are used not only as a component for biological membranes, but also for energy production by fatty acid oxidation (FAO) when other nutrients are depleted [79,83].

It has been suggested that common hexoses other than glucose, such as galactose and fructose, are also involved in the growth and metastasis of cancer cells. Galactose has been shown to induce OXPHOS-dependent metabolism and suppress the growth and metastasis of cancer cells [84,85,86]. On the other hand, fructose has been shown to promote liver metastasis in colon cancer and breast cancer [87]. It has also been shown that aldolase B involved in fructose metabolism promotes the liver metastasis of cancer cells but does not promote metastasis to other organs [87].

### 2.4. Tumor Microenvironments

Stromal cells surrounding tumors, such as endothelial cells, fibroblasts, adipocytes, and immune cells, also support tumor survival and growth [27,55]. It has been shown that these normal cells regulate the growth of cancer cells by secreting growth factors and cytokines, and they actively exchange metabolites, such as lactate and ketone bodies that are generated by aerobic glycolysis, with cancer cells [27,55,88]. Adipocytes also support the survival of cancer cells by supplying fatty acids and glutamine to the tumor microenvironment [77,89].

It is considered that the supply of oxygen and nutrients from existing blood vessels to cancer cells is insufficient for the rapid growth of tumors. Hypoxia induces the reprogramming of glycolysis-dependent metabolism in cancer cells by stabilizing HIFs and increasing the expression of glycolysis-associated genes [37,40]. On the other hand, the chronic deprivation of nutrients induces the metabolism of cancer cells to switch from anabolic to catabolic. For example, metabolic stress activates the p53 and, Liver kinase B1-AMPK cascade, and induces OXPHOS, FAO, and autophagy [90,91]. Although these tumor suppressor genes were originally believed to function in suppressing the cell growth of cancer cells, they are now considered to have a function in preventing cancer cells from undergoing apoptosis during periods of energy deficiency [90,91]. Therefore, it is considered that cancer cells with mutations in these tumor suppressor genes are highly susceptible to the inhibition of specific metabolism pathways and the direct restriction of calories [92,93,94].

Interactions with the extracellular matrix also affect the metabolism of cancer cells. For example, hyaluronic acids in the extracellular matrix induce glycolysis metabolism and epithelial-mesenchymal transition (EMT) in cancer cells through CD44 and receptor tyrosine kinase [95,96,97,98]. On the other hand, it has been shown that mitochondria-related genes, such as the gene for PGC-1α, are upregulated, and OXPHOS is activated in cancer cells that are detached from the extracellular matrix and circulate in blood [60]. The antioxidant system is enhanced in circulating cancer cells in order to remove the increased amounts of ROS resulting from the activation of OXPHOS [99]. In addition, to prevent cell death due to the ROS, some cancer cells create a hypoxic environment by clustering, and induce mitophagy and glycolytic metabolism [84]. Thus, the metabolic reprogramming of cancer cells found in the blood circulation plays an important role in determining the metastatic target and the formation of metastatic foci [100,101,102].

## 3. The Concept of CSCs

Numerous studies have suggested that CSCs are responsible for treatment resistance and cancer recurrence and metastasis [6,7,8,9,10,11,12]. CSCs show very similar phenotypes to normal stem cells, suggesting that there are two possibilities for the development of CSCs, i.e., they are produced due to oncogenic mutations in normal stem cells or progenitor cells with stem cell characteristics, or they are produced when differentiated cancer cells acquire stem cell traits, just like how induced pluripotent stem cells are generated [10]. Like normal stem cells, CSCs usually undergo very slow cell division or remain quiescent in the G0 phase, so they are highly resistant to common anti-cancer drugs that target the vigorous growth of cancer cells [8,10]. In addition, it has been suggested that CSCs contribute to the formation of the heterogeneity in tumors by producing progenitor cells of different differentiation levels via asymmetric division [8,9,10].

There are various methods for identifying and examining CSCs, including: methods for examining the expression of a specific cell surface marker; methods for examining drug resistance ability, such as the function of ABC transporters; methods for measuring specific enzyme activities, such as endogenous protease or ALDH; methods for examining in vitro spheroid-forming ability; and methods for examining tumorigenicity using mouse models [12,103].

CSC traits are controlled by various pluripotent transcription factors, including Oct4, Sox2, Nanog, Klf4, and c-myc, and their upstream signal transduction pathways, including the Wnt, Notch, Hedgehog, Ras/MAPK, Jak/Stat, and PI3K/Akt/mTOR pathways, as well as the characteristics of the complex microenvironment surrounding the tumor, including conditions of hypoxia, stromal cells, growth factors, and the extracellular matrix [6,11,12,97]. In addition to these, it has been revealed in recent years that CSC-specific metabolic mechanisms play an important role in exerting stem cell traits, and they have been attracting attention as new CSC-specific therapeutic targets [15,16,17,18,19,20].

## 4. Metabolism of CSCs

Attempts to identify and inhibit CSC-specific metabolic traits could well lead to the development of therapeutics to eradicate CSCs. To achieve this goal, many researchers are investigating the metabolic phenotypes of CSCs derived from various tissues [15,16,17,18,19,20]. However, there are conflicting hypotheses about whether CSCs depend on glycolysis, mitochondrial metabolism, or other metabolic pathways for maintaining stem cell traits [17,18,20]. In addition, CSC-specific metabolism is responsible for maintaining intracellular redox homeostasis by regulating metabolic balance and synthesizing antioxidants [17,18].

### 4.1. Glycolysis in CSCs

Previous studies have revealed that somatic stem cells, embryonic stem cells, and induced pluripotent stem cells have increased glycolysis activity in order to maintain their stem cell traits [104,105,106,107]. Similarly, in CSCs, such as in breast cancer [108,109,110], lung cancer [111], hepatocellular carcinoma [112,113], glioma [114], ovarian cancer [115], prostate cancer [116], and nasopharyngeal carcinoma [117,118], it has been reported that glycolysis plays an important role in acquiring stem cell properties. For example, CSCs from side-population cells and spheroids overexpress various genes for glycolysis-related factors, such as GLUTs, monocarboxylate transporters, hexokinases, and pyruvate dehydrogenase kinase 1, when compared to non-CSCs [108,110,111,113,118]. In addition, the suppression of the expression or functional inhibition of those glycolysis-related genes markedly reduces the CSC population and attenuates the tumor growth in mouse models [108,109,111,114]. Moreover, the fact that most of the glycolysis-related genes are regulated by HIF-1α could explain why hypoxia stimulation induces CSC traits in differentiated cancer cells in many tumors [114,119,120,121].

Glycolysis-dependent metabolic reprogramming also plays an important role in treatment resistance and metastatic ability, which are important hallmarks of CSCs. Several studies have shown that the inhibition of the glycolysis pathway suppresses the expression of ABC transporter genes in CSCs [122,123]. Nakano et al. have shown that metabolic reprogramming from glycolysis to OXPHOS suppresses the expression of ABC transporter genes by inhibiting the acetylation modification of histone H3 K27 near the promoter of ABC transporter genes [122]. Interestingly, the activation of OXPHOS suppressed the expression of the ABC transporter genes in cells with wild-type p53, while it increased the expression of ABC transporter genes in cancer cells with mutant-p53 [123].

Snail, a key mediator of EMT in cancer cells, is known to induce glycolysis-dependent metabolism by suppressing the expression of the gluconeogenesis-related gene FBP1 in breast cancer [109,124]. On the other hand, recent studies have shown that Snail can switch the metabolic pathway of glucose to PPP by suppressing the expression of phosphofructokinase, platelet [125], suggesting that the role of Snail in glycolysis varies depending on the situation. Moreover, it has been suggested that the switch from OXPHOS to glycolysis-dependent metabolism is required for EMT in several types of cancer cells and for liver metastasis in breast cancer and colorectal cancer cells [87,102,126,127].

### 4.2. Mitochondrial OXPHOS in CSCs

Previous studies have reported that the mitochondria OXPHOS is activated in CSCs in breast cancer [62], lung cancer [128], glioma [129], ovarian cancer [130], pancreatic cancer [131,132], and leukemia [133]. CD133+ cells in pancreatic cancer and glioma have significantly increased gene expression levels of enzymes involved in the TCA cycle and OXPHOS when compared to CD133- cells [129,132]. In many OXPHOS-dependent CSCs, the upregulation of mitochondrial biosynthesis-related genes was observed along with an increase in the numbers and masses of the mitochondria, an increase in the membrane potential of the mitochondria, and the elongation of cristae [62,128,129,130,131,132]. It has been suggested that the characteristics of mitochondria common to CSCs may serve as a new CSC selection marker [134,135,136,137]. Lee et al. have demonstrated that c-myc and mitochondrial protein Mcl1 cooperate to activate OXPHOS in breast cancer CSCs, thereby promoting drug resistance and tumor formation [62]. FAO has also been shown to activate OXPHOS in CSCs by supplying acetyl-CoA in the TCA cycle [89,130,138]. Han et al. demonstrated that carnitine palmitoyltransferase 1A and 2, which are FAO rate-limiting enzymes, enhance the tolerance to radiation of breast cancer cells by enhancing ATP production by FAO [138]. Furthermore, Citrate, which is generated from FAO-derived acetyl-CoA, is oxidized to α-ketoglutarate or pyruvate in cytoplasm and produces cytosolic NADPH [83]. Since PPP-derived NADPH is reduced under glucose-deficient conditions, FAO activation in CSCs can compensate for the lack of NADPH and inhibit ROS-induced apoptosis [83,139].

Whether cancer cells can survive while circulating in the blood is an important factor that determines the success or failure of distant metastasis. Many studies have reported that OXPHOS is activated in circulating cancer cells with the overexpression of PGC-1α [60,100,101]. However, it has been suggested that PGC-1α promotes lung and bone metastasis in breast cancer [59,60,100,101], while suppressing liver metastasis [59].

### 4.3. Metabolic Heterogeneity in CSCs

Even when targeting CSCs derived from the same type of tumor, the observed metabolic phenotypes of CSCs differ in each experiment. Several studies have shown that differences in in vitro and in vivo microenvironments have a significant impact on the metabolic phenotypes of cancer cells [140,141]. The in vivo interactions with the extracellular matrix and the exchange of metabolites and signal transduction substances with stromal cells are important factors for maintaining stem cell traits [6,11,120]. When selecting CSCs from tumors, flow cytometry is often used to analyze the expression of cell surface or endogenous markers, such as CD44, CD133, and ALDH [12]. When cancer cells are taken out from a complicated microenvironment, it is difficult to think that those cancer cells can maintain the metabolic state originally exhibited in the tumor for a long period of time. In addition, conventional cell culture systems use culture media that contain nutrients, such as glucose and glutamine, at levels that exceed those found in physiological conditions, so there is a risk that cancer cells may shift to a metabolic system that depends on those nutrients [140,142]. Therefore, to more accurately elucidate the metabolism of CSCs, it may be necessary to perform quick analyses in fresh cells immediately after they are taken from the body, or to culture CSCs under conditions close to the original environment. Although it is difficult to reproduce the original tumor microenvironment in vitro, organoid culture technology may enable the development of an effective method to reproduce the heterogeneous structure of tumors [143,144,145].

Recent studies have suggested that CSCs contained in the same tumor have non-uniform metabolic phenotypes, and that CSCs with different metabolic phenotypes can coexist (Figure 2) [13,14,132,146]. For example, since many CSCs of pancreatic cancer depend on OXPHOS, OXPHOS-dependent CSCs die when the mitochondria are inhibited [132]. However, in some CSCs, both OXPHOS and glycolysis are activated, so even if OXPHOS is inhibited, glycolysis can be enhanced, and the CSCs can avoid cell death [132]. Luo et al. also showed that there are two types of CSCs in breast cancer, i.e., CD44-positive CSCs and ALDH-positive CSCs [14]. It was shown that in CD44-positive CSCs, the cell cycle is at a quiescent stage and glycolysis is activated, while in ALDH-positive CSCs, cell division proceeds and OXPHOS is activated [13,14]. These results indicate that CSCs do not solely depend on either the glycolysis pathway or the OXPHOS pathway but can flexibly regulate the balance between the strengths and weaknesses of each metabolic pathway depending on the surrounding environment and the stage of differentiation.

### 4.4. Redox Homeostasis in CSCs

To eliminate ROS caused by various internal and external factors, such as increased OXPHOS, radiotherapy, and chemotherapy, CSCs have a potent antioxidant system [14,18,147,148]. The transcription factor Nrf2 is known to be a master regulator of redox homeostasis in normal stem cells and CSCs [14,18,148]. Under basic conditions, ubiquitination by Keap1 suppresses the expression of Nrf2, but ROS stimulation inhibits the interaction between Keap1 and Nrf2, which activates the Nrf2-dependent antioxidant system [149]. In addition, oncogenes, such as Kras and c-myc, have been shown to increase the transcriptional levels of Nrf2 [56,150]. Nrf2 both directly and indirectly regulates the expression of genes related to antioxidant defense, such as the genes for superoxide dismutase, GSH peroxidase, glutamate-cysteine ligase catalytic subunit, glutamate-cysteine ligase modifier subunit, and thioredoxin (Trx), and genes related to drug metabolism, such as the genes for ALDH and ABC transporters [148,149,150,151]. The cysteine/glutamate antiporter (System Xc-) consisting of xCT and 4F2hc plays an important role in the synthesis of GSH, a major intracellular antioxidant [152]. Nrf2 has been shown to promote GSH synthesis by directly controlling the transcription of xCT [153,154]. In addition to activating the antioxidant system, Nrf2 regulates redox homeostasis by reprogramming the metabolism of CSCs [155,156,157,158]. Chang et al. demonstrated that the overexpression of Nrf2 weakens OXPHOS activity and reduces ROS production by inducing the expression of glycolysis-related and non-oxidative PPP-related genes [155]. ROS inhibition by metabolic reprogramming is also observed in other CSCs. It has been shown that the NANOG gene suppresses OXPHOS-derived ROS generation and induces CSC traits by changing the metabolic mechanism of liver cancer CSCs in a glycolysis- and FAO-dependent manner [112]. In addition, Liu et al. have demonstrated that the selective mitophagy of dysfunctional mitochondria induces NANOG gene expression by suppressing p53 gene expression [159].

In contrast, it has been suggested that ROS may promote CSC proliferation and metabolic reprogramming in some tumors [56,62,114,160,161]. Lee et al. demonstrated that c-Myc/Mcl1-overexpressing breast cancer CSCs generate ROS upon the activation of OXPHOS, and that the stabilization of HIF-1α by accumulating ROS is important for maintaining CSC traits [62]. Similarly, in glioma CSCs, the overexpression of HIF-1α associated with increased ROS production was shown to induce CSC traits [114]. Moreover, common to both experiments, the removal of ROS or the suppression of HIF-1α was shown to significantly inhibit CSC traits [62,114]. It is also known that ROS regulate signal transduction that is important for cell proliferation and survival, such as the PI3K/Akt/mTOR, MAPK cascade, Notch, and Wnt/b-catenin pathways [56,161].

On the other hand, it has been suggested that the role of ROS differ depending on the stage of cell differentiation (Figure 3). Normal hematopoietic stem cells (HSCs) in a dormant state have reduced ROS production due to the activation of glycolysis and suppression of OXPHOS [105,161,162]. When it becomes necessary to supplement the blood, the HSCs activate OXPHOS with ROS production to promote cell growth [105,161,162]. Therefore, artificially inhibiting glycolysis or inducing the production of ROS reduces the number of HSCs in the dormant state [105,161,162]. OXPHOS-dependent metabolic reprogramming has also been observed during normal tissue regeneration [163]. Similar to normal tissues and stem cells, when ROS production is increased, the number of resting cells decrease and cell division is promoted among CSCs in leukemia [133], breast cancer [14], and head and neck cancer [155]. Lagadinou et al. revealed that quiescent leukemia stem cells overexpress Bcl-2 to suppress ROS production [133]. It has been shown that the inhibition of Bcl-2 selectively eliminates ROS-low quiescent leukemia stem cells but has little effect on ROS-high differentiated leukemia cells [133]. These results strongly suggest that for the tumor to maintain the CSC population, it is necessary to maintain an appropriate amount of ROS by modulating the balance of metabolic activity and the antioxidant system.

## 5. Therapeutic Strategies for Targeting CSC Metabolism

Basically, dormant CSCs are considered to have strong resistance to conventional chemotherapy and radiation therapy that target active cell proliferation [8,10]. However, a treatment targeting CSC-specific metabolism may be able to selectively eliminate even dormant CSCs by depleting the energy or metabolites required for the survival of CSCs [15,16,17,18,19,20]. In addition, the inhibition of CSC-specific metabolism has been reported to increase the sensitivity to conventional chemotherapy [15,16,17,18,19,20]. Furthermore, since the importance of CSC-specific metabolism has been clarified by previous studies, it has been suggested that not only conventional chemotherapy, but also diet therapy is important in reducing the risk of cancer malignancy and a poor prognosis [20].

The mitochondrial metabolism of CSCs is currently attracting attention as an effective therapeutic target [17,19,20,50,55]. In addition to OXPHOS, mitochondria are the center of various metabolic pathways that provide essential cellular components, such as amino acids, including glutamine, fatty acids, and ketone bodies through the TCA cycle [50,51,52,55]. It has also been shown that mitochondria function normally even in glycolysis-dependent cancer cells, and that glutamine metabolism and FAO are activated [50,51,52,54,55]. In vitro and in vivo experiments have shown that inhibitors of mitochondrial OXPHOS, such as metformin, phenformin, and menadione, suppress CSC traits [132,163,164,165,166,167]. Also, since mitochondria are derived from a prokaryote that originally parasitized eukaryotic cells, it is known that its function can be inhibited by treatment with various antibiotics [17,18,19,20,168,169,170]. It has been shown that tetracycline antibiotics, such as doxycycline, specifically inhibit CSC traits by inhibiting mitochondrial biosynthesis, but they do not exert toxicity on normal cells or the bulk of the cancer [168,169,170]. On the other hand, the inhibition of glycolysis has been shown to be effective against some CSCs, but in many cases, it induces the resistance of cancer cells to therapy [108,110,171].

The metabolic state of CSCs in tumors is very heterogeneous, and the metabolism of each CSC is flexible enough to adapt to metabolic changes. Therefore, in most cases, even if a single metabolic pathway is inhibited, it is extremely difficult to uniformly eliminate all CSCs. However, the simultaneous inhibition of multiple metabolic pathways may prevent resistance acquisition and eliminate CSCs more efficiently (Figure 4). Since metformin is already approved by the U.S. Food and Drug Administration (FDA) as an anti-diabetic drug, its combined effect with a wide range of metabolism inhibitors is being tested [68,132,172,173,174,175]. In addition to the ETS, metformin is known to inhibit mTORC1 activity indirectly or directly [176,177]. mTORC1 has a central role in the metabolic regulation of cancer cells, so metformin may enhance the effects of other metabolic inhibitors by inhibiting the metabolic reprograming of CSCs. The glutaminase inhibitors BPTES and CB839 strongly inhibit the growth of glutamine-addicted tumors even when used alone, but they have been shown to be more potent when combined with other metabolic inhibitors or conventional chemotherapy [68,69,70,171,178,179]. The FAO inhibitor etomoxir also enhanced the therapeutic effect of conventional chemotherapy in several types of cancers in vivo [138,180,181]. The selective Nrf2 inhibitor trigonelline, the xCT inhibitor sulfasalazine, the GSH inhibitor buthionine sulfoximine, and the Trx inhibitor auranofin induce cell death by accumulating ROS in CSCs. When the GSH pathway is inhibited, Trx expression is increased to compensate for it, indicating that the inhibition of both GSH and Trx at the same time can eliminate CSCs more efficiently than the inhibition of GSH or Trx alone [182,183,184,185].

It is known that the activation of aerobic glycolysis in cancer cells leads to glucose depletion in the microenvironment and acidification due to lactate over-secretion, thereby impairing immune cell function [33,34,186]. In contrast, the activation of mitochondrial OXPHOS in cancer cells may promote immune responses [187]. Harel et al. conducted a detailed analysis of the metabolic phenotypes of melanoma patients who underwent cancer immunotherapy, and the tumors of patients showing a therapeutic effect had significantly increased expression levels of genes related to mitochondrial OXPHOS when compared to those of the non-responders [187]. Furthermore, they found that the expression of OXPHOS-related genes restored the antigen-presenting function of cancer cells [187]. Therefore, glycolysis inhibition or OXPHOS activation can be expected to enhance the effect of cancer immunotherapy by simultaneously activating the antigen-presenting function of cancer cells and the function of immune cells [187,188].

Previous studies have suggested that calorie restriction and diet therapy may reduce the incidence and progression of cancer [189]. However, little is known about the effects of these external factors on CSCs. At the least, experiments using model mice have suggested that overeating after calorie restriction promotes the proliferation of CSCs and distant metastases [190]. On the other hand, some food-derived ingredients, e.g., eicosapentaenoic acids and docosahexaenoic acids, which are omega-3 polyunsaturated fatty acids, and vitamin C, have a therapeutic effect on CSCs [191,192,193,194]. However, there have been no reported clinical cases in which these components were effective for tumor regression. Ketogenic diets are expected to reduce the risk of the malignant transformation of cancer, but conflicting results on the suppression/promotion of tumor growth have been obtained even within the same tumor types [195]. In addition, Martinez-Outschoorn et al. reported that ketone bodies induce stemness properties in breast cancer cells [196]. Similar to how cancer cells acquire resistance to certain metabolic inhibitors, the continual intake of biased diets may induce CSCs to adapt to the biased environment, which may increase the risk of cancer malignancy.

Both in vitro and in vivo experiments have demonstrated that the inhibition of metabolism may induce a reduction in the CSC population and tumor regression, but many of the metabolic pathways activated in cancer cells are also activated in various tissues, organs, normal stem cells, and the like. Therefore, as with general anticancer agents, many metabolic inhibitors have serious side effects, such as acidosis by the treatment of anti-diabetic drugs [94,197,198]. To reduce the side effects, it is necessary to accurately determine the effects of the metabolic inhibitors on cancer cells and normal cells, and to identify and establish highly selective methods of eliminating cancer cells.

## 6. Conclusions

In light of the technological progress and great efforts of researchers in recent years, the importance of CSCs and their metabolic regulation that underlie tumor heterogeneity have gradually been revealed. Numerous studies have revealed that each CSC has a distinct metabolic mechanism that differs from that of non-CSCs, and CSC-specific metabolism is therefore an attractive target for cancer therapy. However, the simultaneous presence of multiple CSCs with distinct metabolic phenotypes, even within same tumor types, and the metabolic flexibility of CSCs make the development of effective treatments difficult. If the various metabolic pathways are obstructed in a blind way, it causes serious side effects in normal tissues. As such, it is necessary to efficiently identify and target the specific metabolic vulnerabilities of each type of CSC. Although there are critical factors that play central roles, the activity of each metabolic pathway is regulated by complex interactions between many factors. For example, even if we focus only on glycolysis, the pathway is not isolated, and at each reaction step, different enzymes regulate the reactions and the metabolites may flow into or out of different metabolic pathways. Within such a complicated metabolic system, it is very difficult to identify potential targets in CSCs. However, recent studies have attempted to identify such targets by systematically linking the metabolic profiles of tumors with genome-wide transcriptome and proteomics analysis [142,199,200,201]. In the near future, by accumulating such research data, it is expected that the identification of the metabolic vulnerabilities of each tumor will be possible, which will enable the development of more efficient diagnostic techniques and therapies.

## Figures and Tables

**Figure 1 cancers-12-02780-f001:**
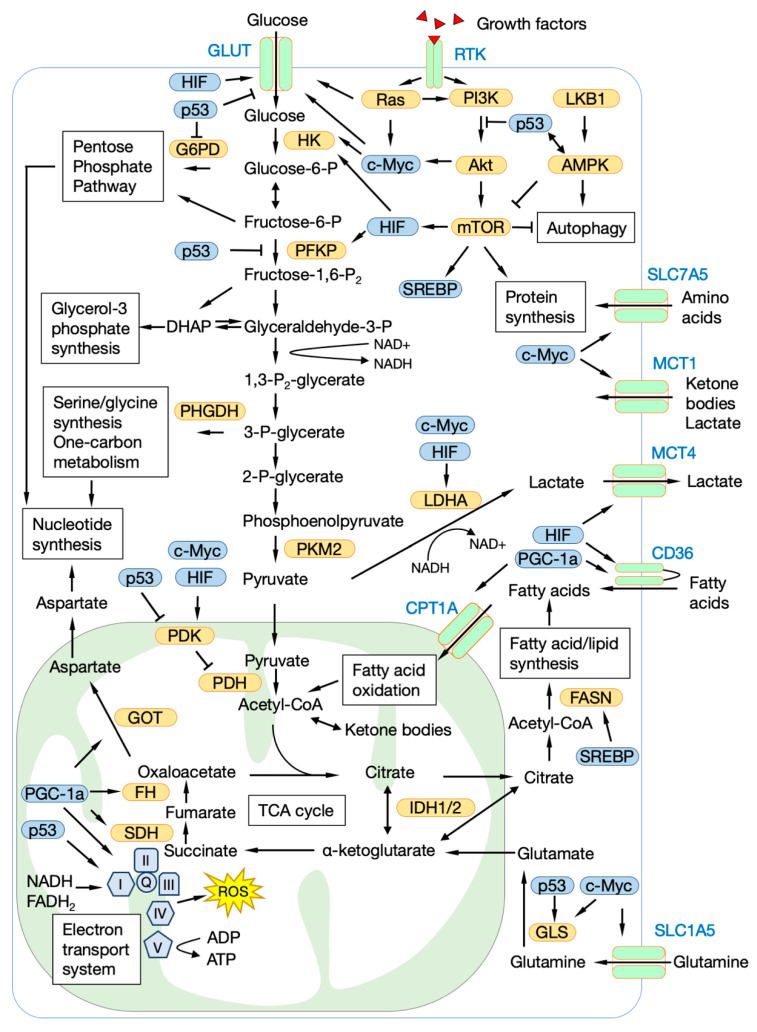
The central carbon metabolism map in cancer cells. GLUT, glucose transporter; RTK, receptor tyrosine kinase; MCT, monocarboxylate transporter; SLC7A5, solute carrier family 7 member 5; SLC1A5, solute carrier family 1 member 5; HK, hexokinase; G6PD, glucose-6-phosphate dehydrogenase; PFKP, phosphofructokinase platelet; PHGDH, phosphoglycerate dehydrogenase; PKM2, pyruvate kinase M2; LDHA, lactate dehydrogenase A; PI3K, phosphatidylinositol-3 kinase; LKB1, Liver kinase B1; AMPK, AMP-activated protein kinase; mTOR, mammalian target of rapamycin; FASN, fatty acid synthase; GLS, glutaminase; PDK1, pyruvate dehydrogenase kinase 1; PDH, pyruvate dehydrogenase; IDH, isocitrate dehydrogenase; SDH, succinate dehydrogenase; FH, fumarate hydratase; GOT, glutamic-oxaloacetic transaminase; I-X, electron transport chain complex I-X; PGC-1a, peroxisome proliferator-activated receptor gamma coactivator 1-alpha; HIF, hypoxia inducible factor; SREBP, sterol regulatory element binding transcription factor.

**Figure 2 cancers-12-02780-f002:**
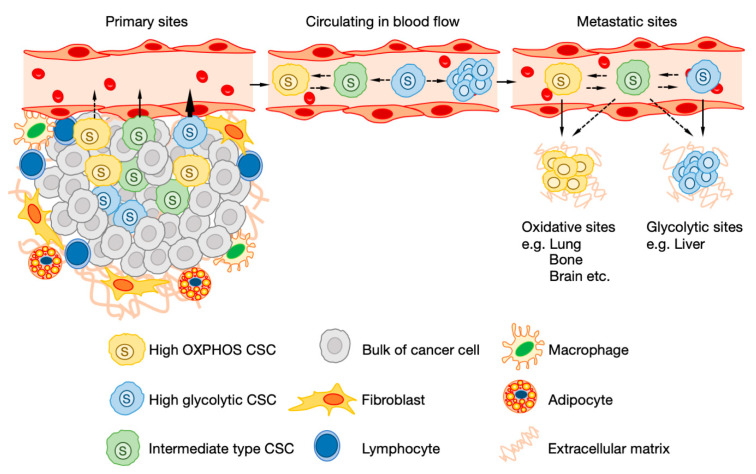
The metabolic heterogeneity and flexibility of cancer stem cells (CSCs). The metabolic phenotypes of each CSC are very heterogeneous even within the same tumor. High glycolytic CSCs have more invasive ability compared to high oxidative phosphorylation (OXPHOS) CSCs. Circulating cancer stem cells are dependent on OXPHOS or have a higher metabolic capacity. Glycolytic CSCs reprogram metabolism to OXPHOS-dependent manner or maintains glycolytic metabolism by forming clusters during circulating. Eventually, CSCs metastasize to organs that are compatible with metabolic phenotypes.

**Figure 3 cancers-12-02780-f003:**
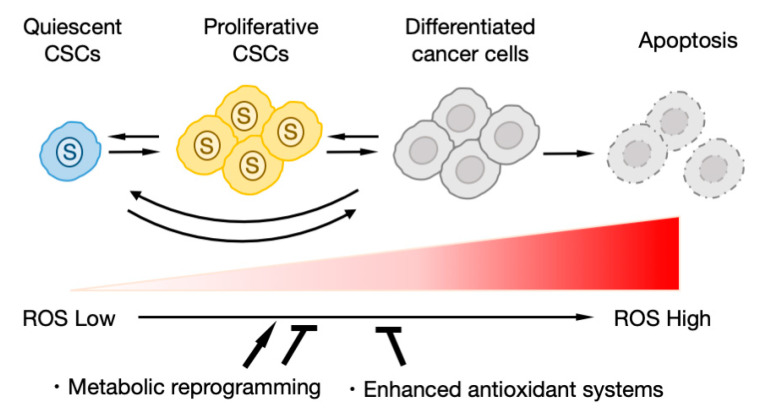
The redox regulation and differentiation of CSCs. Quiescent CSCs maintain low-level reactive oxygen species (ROS) by glycolysis-dependent metabolism or enhanced antioxidant systems. ROS accumulation associated with OXPHOS activation induces CSC proliferation and differentiation.

**Figure 4 cancers-12-02780-f004:**
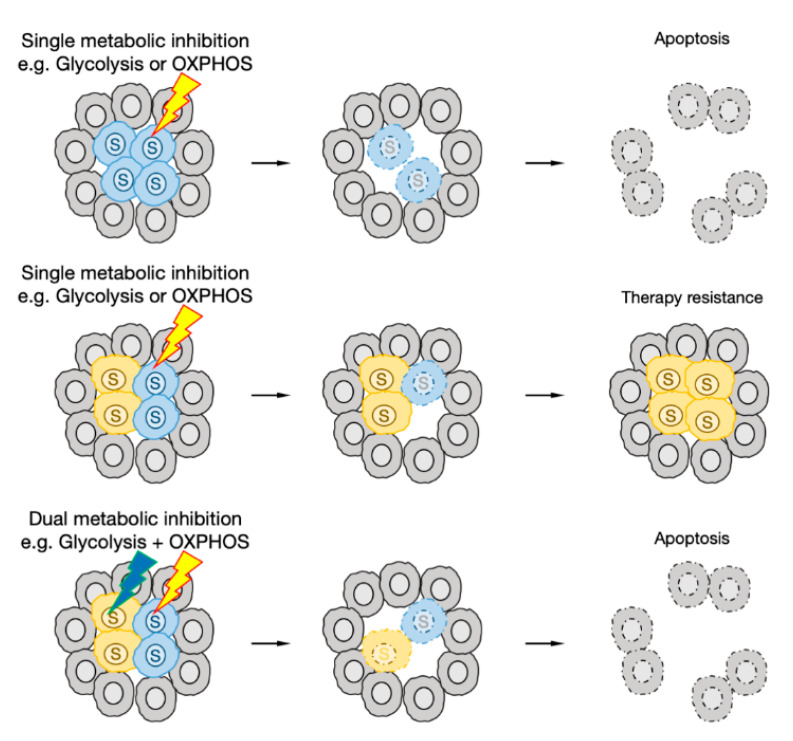
Therapeutic strategies targeting the metabolism of CSCs. If CSCs depend on a particular metabolism, a single metabolic inhibitor can effectively eliminate the tumor. When multiple CSCs with distinct metabolic phenotypes are present in the same tumor, it is necessary to simultaneously inhibit multiple metabolic pathways to eliminate the tumor.

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
