# Peer review of "The Metabolic Heterogeneity and Flexibility of Cancer Stem Cells"

_cancers, 2020, doi:10.3390/cancers12102780_

Round 1
Reviewer 1 Report
In the present manuscript by Tanabe and Sahara, the authors provide a review of the current literature surrounding the metabolism of cancer stem cells (CSCs). They provide an overview of central carbon metabolism in cancer, then demonstrate how it is altered in cancer stem cells and finally highlight the known approaches for targeting metabolism in CSCs. The insights into dual metabolic strategies for therapeutic intervention, as well as their caveats, are particularly important to discuss in this rapidly evolving research field. The manuscript is generally well written, provides a very thorough review of the current literature and cites appropriate primary and review articles.
Comment 1 - References need to be double checked. References somewhere around 65-77 seem to be off by up to five places. For example, ref #70 on line 144 should be ref #75.
Comment 2 - There are a lot of Figures (5) and some are not all that useful or are relatively simple concepts. This is particularly true for Figure 2 and 4. Consider removing Figure 2 or adding a novel angle to the figure. Also, how is the PPP changed between these three contexts?
Very minor comments:
Figure 1 – fix 3PG spelling.
Line 307 – fix grammar.
Line 444 – remove the second “in recent years”.
Author Response
Response to #1 reviewers’ comments
Comment 1 - References need to be double checked. References somewhere around 65-77 seem to be off by up to five places. For example, ref #70 on line 144 should be ref #75.
Response to reviewers’ comment 1:
Thanks for pointing it out. We have fixed reference numbers as follows:
Line 154: [70]→[75], line 155: [71]→[76], line 157: [65,66,71,72]→[65,66,76,77], line 182: [77,88]→[76,88]
Comment 2 - There are a lot of Figures (5) and some are not all that useful or are relatively simple concepts. This is particularly true for Figure 2 and 4. Consider removing Figure 2 or adding a novel angle to the figure. Also, how is the PPP changed between these three contexts?
Response to reviewers’ comment 2:
Thanks for the comment and helpful suggestion. According to the reviewer's suggestion, we have removed Figure 2. Also, we could not find any clear evidence that PPP activity is changed by the metabolic phenotype of CSCs. However, we think there is no doubt that the PPP activity is upregulated in most cancer cells.
Very minor comments:
Figure 1 – fix 3PG spelling.
Response to reviewers’ comment 3:
Thanks for pointing it out. We have fixed 3PG spelling.
Line 307 – fix grammar.
Response to reviewers’ comment 4:
Thanks for pointing it out. We have fixed grammar in Figure 3 legend as follows:
Line 333:
The metabolic phenotype of each CSCs is very heterogeneity even the same tumor.
→The metabolic phenotypes of each CSC is very heterogeneity even within the same tumor.
Line 444 – remove the second “in recent years”.
Response to reviewers’ comment 5:
Thanks for pointing it out. We have removed the second “in recent years” (line 476).
Reviewer 2 Report
This manuscript by Tanabe et al. is described from the metabolic heterogeneity and flexibility in cancer stem cells (CSC) down to the basis of metabolism. This is a well-written review of interest to the field of basic and clinical cancer research. The manuscript is well organized, well written, and deals with the dynamics of the metabolic state of CSC from a different, original, and updated angle. Therefore, this manuscript is suitable for publication in "Cancers".
Author Response
We would like to thank you for your comments.
The English for this revise was proofread by native foreigner.
Reviewer 3 Report
The review article by Tanabe and Sahara, addresses the metabolic heterogeneity of cancer stem cells and highlights the challenges in targeting these cells in light of cancer therapy. Overall the article is well written and I have only some minor comments that the authors might want to consider.
Line 59:
While it is correct that conversion of glucose to pyruvate comprises 10 enzymes, the total number of reactions (or „steps“) is 17, because the number of reactions after the aldolase step (split of C6 body into 2 C3 bodies) is duplicated. Thus, I would either use “enzyme” instead of “step” or adapt the number of reactions.
Line 61:
Can the authors please explain how they get to the number of 36 ATP as the net production from glucose? According to Biochemistry text book (Lehninger) it should be 30 ATP (or 32 depending on the transport system to import NADH into the mitochondria). - Assuming 1 NADH= 2.5 ATP and 1 FADH= 1.5 ATP-.
Line64:
Even though we read quite commonly about the Warburg effect, lactate release and support of biosynthetic routes, it is however, technically not fully correct because whenever lactate is made and excreted out of the cells, the carbon is lost and can no longer be used to fuel PPP, serine or glycerol (lipid) synthesis. This, discrepancy has also been discussed in other recent reviews (DOI: 10.1038/s42255-020-0172-2). I would consider to slightly rephrase this section.
Figure 1
Typo in 3-P-glyceralte
Line105/106
Shouldn‘t the enzyme be glutamate-oxaloacetate transaminase?
Line 267-269:
It is not clear to me how FAO (generating NADH) can compensate for a loss of PPP derived NADPH? Maybe this can be further explained in the text.
Author Response
#3 Reviewer's comments
Line 59:
While it is correct that conversion of glucose to pyruvate comprises 10 enzymes, the total number of reactions (or „steps“) is 17, because the number of reactions after the aldolase step (split of C6 body into 2 C3 bodies) is duplicated. Thus, I would either use “enzyme” instead of “step” or adapt the number of reactions.
Response to reviewers’ comment 1:
Thanks for the comment and helpful suggestion. According to the reviewer's suggestion, we have fixed the text as follows:
Line 66: [in 10 steps]→[by 10 enzymes]
Line 485: [and at each of the 10 steps]→[and at each reaction step]
Line 61:
Can the authors please explain how they get to the number of 36 ATP as the net production from glucose? According to Biochemistry text book (Lehninger) it should be 30 ATP (or 32 depending on the transport system to import NADH into the mitochondria). - Assuming 1 NADH= 2.5 ATP and 1 FADH= 1.5 ATP-.
Response to reviewers’ comment 2:
Thanks for pointing it out. We have fixed the number of ATP (line 68).
Line 64:
Even though we read quite commonly about the Warburg effect, lactate release and support of biosynthetic routes, it is however, technically not fully correct because whenever lactate is made and excreted out of the cells, the carbon is lost and can no longer be used to fuel PPP, serine or glycerol (lipid) synthesis. This, discrepancy has also been discussed in other recent reviews (DOI: 10.1038/s42255-020-0172-2). I would consider to slightly rephrase this section.
Response to reviewers’ comment 3:
Thanks for the comment and helpful suggestion. According to the reviewer's suggestion, we have rephrased this section as follows:
Line 71/72:
because glycolysis is linked to various other metabolic pathways that produce biogenic substances required for cell growth.
→because glycolytic intermediates, which are generated in the pre-stage of lactate secretion, are linked to various other metabolic pathways that produce biogenic substances required for cell growth.
Figure 1:
Typo in 3-P-glyceralte
Response to reviewers’ comment 4:
Thanks for pointing it out. We have fixed the typo (Figure 1).
Line105/106:
Shouldn‘t the enzyme be glutamate-oxaloacetate transaminase?
Response to reviewers’ comment 5:
Thanks for pointing it out. We have fixed the enzyme name (line 118/119).
Line 267-269:
It is not clear to me how FAO (generating NADH) can compensate for a loss of PPP derived NADPH? Maybe this can be further explained in the text.
Response to reviewers’ comment 6:
Thanks for the comment and helpful suggestion. According to the reviewer's suggestion, we have added the explanation in the text as follows:
Line 291/292:
Citrate, which is generated from FAO-derived acetyl-CoA, is oxidized to α-ketoglutarate or pyruvate in cytoplasm, and produces cytosolic NADPH.